# Offshore Platform Extraction Using RadarSat-2 SAR Imagery: A Two-Parameter CFAR Method Based on Maximum Entropy

**DOI:** 10.3390/e21060556

**Published:** 2019-06-02

**Authors:** Qi Wang, Jing Zhang, Fenzhen Su

**Affiliations:** 1State Key Laboratory of Resources and Environmental Information System, Institute of Geographical Sciences and Natural Resources Research, CAS, Beijing 100101, China; 2Faculty of Geomatics, Lanzhou Jiaotong University, Lanzhou 730070, China; 3Gansu Provincial Engineering Laboratory for National Geographic State Monitoring, 88 Anning Rd., Lanzhou 730070, China

**Keywords:** offshore platform extraction, two-parameter CFAR, maximum entropy, Pearl River Estuary Basin

## Abstract

The ability to determine the number and location of offshore platforms is of great significance for offshore oil spill monitoring and offshore oil and gas development. Considering the problem that the detection threshold parameters of the two-parameter constant false alarm rate (CFAR) algorithm require manual and repeated adjustment of the during the extraction of offshore platform targets, this paper proposes a two-parameter CFAR target detection method based on maximum entropy based on information entropy theory. First, a series of threshold parameters are obtained using the two-parameter CFAR algorithm for target detection. Then, according to the maximum entropy principle, the optimal threshold is estimated to obtain the target detection results of the possible offshore platform. Finally, the neighborhood analysis method is used to eliminate false alarm targets such as ships, and the final target of the offshore platform is obtained. In this study, we conducted offshore platform extraction experiments and an accuracy evaluation using data from the Pearl River Estuary Basin of the South China Sea. The results show that the proposed method for platform extraction achieves an accuracy rate of 97.5% and obtains the ideal offshore platform distribution information. Thus, the proposed method can objectively obtain the optimal target detection threshold parameters, greatly reduce the influence of subjective parameter setting on the extraction results during the target detection process and effectively extract offshore platform targets.

## 1. Introduction

Petroleum and natural gas resources are the economic lifeblood of national development and occupy an extremely important strategic position. As offshore drilling technology has matured, the exploitation of offshore oil and gas resources has increased. As a primary type of equipment for offshore oil and gas resource development, the number and location of offshore platforms reflect a country’s oil and gas resource development intensity and capacity to some extent [1]. In addition, information concerning the distribution of offshore platforms is also of great significance in applications such as maintaining channel security and monitoring offshore oil spills [2].

Historically, offshore platform distributions were obtained via field surveys, which were time-consuming and costly. Remote sensing images cover a wide area, making them convenient for providing information on large numbers of offshore platforms. Currently, many studies have applied remote sensing methods to extract information about offshore platforms. For example, Xing [2], and Liu et al. [3] studied the application of offshore platform extraction using Landsat satellite imagery data. As Landsat image data are greatly affected by the marine meteorological environment, a time-series strategy is usually adopted to overcome the impact, but it also greatly increases the demand for data volume. Because the offshore platform production process is usually accompanied by the combustion of exhaust gas, in the night shows a bright firelight. Therefore, there are also studies using night light image data for the detection of offshore platforms [3] and the estimation of combustion volume [4]. Again, this data is susceptible to cloud cover, especially in equatorial regions. Moreover, due to the limitation of spatial resolution, the quantity and position accuracy of offshore platform extraction is relatively low. Relative to the above two types of data, Synthetic Aperture Radar (SAR) data is less affected by marine weather, light and other conditions, and can observe offshore targets in a macro, long-term, continuous, dynamic and real-time manner, especially in the detection of ships, offshore platforms and other targets, which are widely used and have high extraction accuracy [5,6]. At present, there has been a lot of analysis and research on offshore target detection of SAR images [7,8,9,10], and SAR images have been the main data for the application of offshore target extraction.

At present, many methods exist for detecting targets at sea using SAR images; among these, the most representative methods are Otsu [11,12], constant false alarm rate (CFAR) [13] and wavelet analysis [14]. Of these representative detection methods, the CFAR algorithm is the most commonly used radar signal detection algorithm for detecting targets from SAR images [15,16]. However, there are still many problems in the specific research and application of CFAR algorithm. Therefore, in the follow-up research, many researchers have made continuous research and improvement on the basis of this method. For instance, the CFAR algorithm uses the background mean value to represent the cluttered area in the background window [17,18]. This approach is effective when the background clutter is simple, but when the clutter is complex, a single parameter is insufficient to fully describe the clutter. Therefore, Novak proposed a two-parameter CFAR detection algorithm based on the original one for more complex clutter. The algorithm uses the mean and variance of the pixel intensity in the background window to describe the clutter, an approach that has strong adaptability in more complex clutter environments [19]. Subsequently, Ai et al. [20] further improved the two-parameter CFAR algorithm to overcome its tendency to fail in detecting relatively close targets. However, this algorithm sets only the target and background windows. Because target pixels may exist in the background window as the window slides, this method removes the target portion in the background window; then, it counts the remaining pixels to obtain the estimated clutter mean and variance. However, the simulation results show that although this method does not miss near-target detections, false target detections can still occur. In addition, some researchers have also conducted a series of studies on the detection threshold parameters, the computational efficiency and the clutter estimation methods of the two-parameter CFAR algorithm. When Liang et al. applied the two-parameter CFAR algorithm to extract targets, the focus was on determining the appropriate detection parameters for the two-parameter CFAR. This method estimated the false alarm rate coefficient according to the cluster distribution of the image to ensure that all the targets of the offshore platform were detected when the false positive rate and the false negative rate were relatively low [21]. Xu et al. [22] targeted the shortcomings of the two-parameter CFAR algorithm in terms of its computational efficiency; to improve the computational efficiency they adjusted the detection window and the moving step size. When using target window detection, the value of the false alarm rate coefficient t is set several times to obtain the optimal detection threshold. Greidanus et al. proposed a pixel-based CFAR multi-view radar image detector. The method assumes that the clutter background has a K-distribution and corrects it based on the deviation between the actual sea clutter and the K-distribution, achieving a fast and robust clutter background estimation method. Then, the target is detected by setting the appropriate false alarm rate control coefficient. Their testing demonstrated that the algorithm performs well across a wide range of SAR environments [23]. However, all the above methods have a common deficiency—the false alarm rate control coefficient t needs to be adjusted repeatedly based on the clutter model to achieve an optimal detection effect. Especially in offshore target extraction applications, the control coefficient of false alarm rate is typically set based on operator judgment from previous experience or through repeated tests. This requirement makes applying this method tedious and subjective and has a large impact on the detection results.

To solve the above problems, this paper proposes a two-parameter CFAR algorithm based on entropy theory, which can objectively obtain the optimal false alarm rate control coefficient. First, a series of threshold parameters are obtained using the two-parameter CFAR algorithm for target detection. Then, according to the maximum entropy principle, the optimal threshold is estimated to obtain the detection results of the possible offshore platform targets. Finally, a neighborhood analysis method is used to eliminate false alarm targets such as ships based on the stability characteristics of the platform target, and the final targets of the offshore platform are obtained. The remainder of this paper is organized as follows. Section 2 introduces the case study area and the data set used. Section 3 describes the principle and detection steps of the proposed method. Section 4 presents the extraction results of the oil and gas platform and the accuracy evaluation and analyses the results of an image comparison performed for verification purposes. Section 5 compares the advantages of the proposed method for offshore platform extraction with the approaches in previous studies, lists some shortcomings of this method and discusses the impact of other maritime targets on the extraction results.

## 2. Study Areas and Datasets

### 2.1. Study Areas

For this study, we selected the Pearl River Estuary Basin in the northern part of the South China Sea (Figure 1a,b, 111°26′58.665″E, 19°16′29.833″N–117°19′22.672″E, 22°12′1.486″N, 114°44′41.906″E, 19°20′31.94″N–116°40′22.305″E, 22°27′29.562″N) as the study area. The study area is in China’s main offshore oil and gas resource producing region and features a high degree of oil and gas exploration, large oil and gas resource output and dense distribution of offshore platforms that is representative of the region. In addition, in from the existing archived image data, the image coverage of the selected study area is relatively complete.

### 2.2. Datasets

The RadarSat-2 SAR georeferenced fine resolution (SAR georeferenced fine resolution, SGF) images (Table 1, Figure 3a) of the 2-width scan (ScanSAR Wide) mode was collected for offshore platform extraction. This image product underwent processing such as ground distance conversion, multi-view processing, and image calibration. The dates of the two images are 2014-03-09 and 2014-10-12, and the time interval is 7 months, which is beneficial to target detection during the two phases of the neighborhood analysis to eliminate the moving ship target. The images have a spatial resolution of 100 m, VH polarization and a width of 500 km × 500 km. In general, the co-polarization (HH, VV) mode is more favorable for a target whose structure is parallel to the radar’s viewing direction, and cross polarization (HV, VH) better detects targets at a certain angle from the radar’s viewing angle [24]. The deck heights of main offshore platform structures are generally located approximately 20 m above the water surface and form a certain angle with the radar viewing direction. Therefore, the selection of VH-polarized data is more conducive to target detection (Figure 2).

GF-1 optical images collected near the same dates acquired from the China Center for Resources Satellite Data and Application (CRESDA, http://www.cresda.com/CN/) website were used for comparative verification and analysis of the extraction results. The spatial resolution of the GF-1 images is 2 m. Marine environments are greatly affected by clouds and rainy weather, which have certain influences on the imaging quality of optical images. When selecting the images, we controlled the cloud coverage rate to be <20%. The list of images and their coverage are shown in Table 2 and Figure 3b.

In addition, the study also collected the actual spatial distribution of offshore platforms and related auxiliary data in the study area through a field survey, including (1) Petroliferous basin data, referenced to the *Atlas of China’s Petroliferous Basins*; (2) vector data such as national boundaries, provided by the global administrative division website (http://www.nhjd.net/); (3) field survey data of the number and distribution of offshore platforms in the Pearl River Estuary Basin in June 2014 (no new or abandoned offshore platforms were found in the Pearl River Estuary Basin from March 2014 to October 2014).

## 3. Methodology

The offshore platform detection method proposed in this paper includes three main steps, as follows:(1)Image preprocessing. The preprocessing of the two-phase SAR images used in this experiment included geometric correction and radiometric correction, landmasking of the image research area and image-filtering.(2)Offshore platform detection. The pre-processed two-phase images were processed using the two-parameter CFAR detection program based on maximum entropy, including target detection and detection threshold calculation. The program outputs a possible offshore platform target set, including both offshore platforms and ships.(3)Neighborhood analysis. The offshore platform targets are determined by comparing the position-invariant characteristic of offshore platforms with the moving characteristics of ships.

The detailed steps are shown in Figure 4:

### 3.1. Image Preprocessing

#### 3.1.1. Image Correction

The RadarSat-2 images used in this paper are SAR georeferenced fine resolution (SGF) products. Before using them to extract offshore platforms, the images must undergo a series of image-correction operations, including geometric correction and radiation correction. We calibrated the data using the SARscape module in ENVI 5.3 software (Beijing Lan Yu Fangyuan Information Technology Co. Ltd., Beijing, China). Because selecting control points at sea is difficult, this study established RPC files based on the ephemeris parameters of the data to perform the geometric correction. To correct the radiation, the images are corrected according to image metadata files.

#### 3.1.2. Mask Processing

Offshore platforms are mostly located in sea areas far from shore. These offshore areas have large numbers of complex surface features, including many artificial metal manufacturing facilities with a strong backward scattering coefficient. The coverage range of the RadarSat-2 data used in this study extends from well out at sea to near-shore areas; these nearshore facilities can impact the extraction results. Therefore, the images were masked before beginning the offshore platform extraction process (Figure 5).

#### 3.1.3. Filtering

Due to the special coherent imaging mechanism of SAR, after imaging, a large amount of coherent speckle noise remains in the image that seriously impacts the subsequent target detection [25]. To improve the interpretability of ground objects in SAR images, this speckle noise must be filtered to effectively suppress the noise while retaining as many of the target’s edge details as possible. A comparison among many filtering methods found that sigma filtering is a better SAR image filtering method [26,27]. In the sigma filtering algorithm, the degree of filtering of SAR images can be controlled by setting the threshold K and the size of the filtering window [19]. After several tests, we obtained the best effect when the sigma filtering window is set to 3 × 3 and the threshold K is set to 8 (Figure 6).

### 3.2. A Two-Parameter CFAR Target Detection Method Based on Maximum Entropy

#### 3.2.1. Two-Parameter CFAR Target Detection Algorithm

The complex and variable sea conditions and the imaging characteristics of SAR images themselves make SAR images highly noisy (Figure 7), which causes a large number of false alarm targets during target detection and results in low accuracy of the automatic detection results [28]. Although the K-CFAR algorithm can provide accurate background sea clutter distribution model, as a global threshold detection algorithm, it is generally applicable to images with a uniform background. Therefore, it is not possible to detect ocean targets using a global threshold; consequently, detection algorithms with local adaptive ability should be applied. The two-parameter CFAR detection algorithm, which is based on the assumption that the background clutter is a Gaussian distribution [29], uses a local window for target detection that can adapt to changes in the background clutter. This approach can maintain good detection performance even in more complicated clutter environments, [30,31,32,33].

The two-parameter CFAR detection algorithm needs to create three detection windows centered on each pixel during target detection: a background window $W_{b}$, a protection window $W_{p}$ and a target window $W_{t}$ (shown in Figure 8). The background window represents the statistical information of the sea clutter background. The protection window helps ensure that the target pixel does not leak into the background window. The target window is mainly used to detect offshore platforms and ship targets.

Generally, the size of the target window is set to the size of the largest target in the detected image; the size of the protection window is twice that of the target window; and the background window is set to the size of the protection window + *2n*, where the value of *n* is usually 3 [30]. Because the directions of offshore platforms and ship targets are not fixed, the detection window is a square. The two-parameter CFAR algorithm performs target detection by calculating the mean $\mu_{b}$ of all the pixels in the background window and the background standard deviation $\sigma_{b}$. If the grey value of a pixel in the target window is $I_{t}$, the criterion for determining whether a pixel in the target window is a target is [20]:(1)$$I_{t}\;\begin{array}{c} >\\ \leq \end{array}\;\mu_{b}+\sigma_{b}\cdot t$$

When $I_{t}>\mu_{b}+\sigma_{b}\cdot t$, the pixel is a considered target. Otherwise, it is considered to be background. Here, *t* is a standardized factor and is usually a constant (also known as the constant false alarm rate control coefficient) that controls the false alarm rate [34,35].

Here, we transform formula (1) to obtain a function ($F\left(I_{t},\mu_{b},\sigma_{b}\right)$) of the grayscale value($I_{t}$) of pixels in the target window, and the mean($\mu_{b}$) and standard deviation($\sigma_{b}$) of the background window. The result of this function can be understood as a set of coefficients *T*. In fact, the target detection process is to determine whether the pixel is a target by comparing the numerical value between the coefficient *T* and the constant false alarm rate control coefficient *t*, as shown in Equation (2).
(2)$$F\left(I_{t},\mu_{b},\sigma_{b}\right)=\frac{I_{t}-\mu_{b}}{\sigma_{b}}=T\;\begin{array}{c} >\\ \leq \end{array}\;t$$

Applying Equation (2) to the SAR image in Figure 9b obtains the distribution of the coefficient *T* with respect to the image gradation (Figure 9c). Because of the window calculation with local adaptive ability, there is obvious visual separability between the target and the background in the distribution image of coefficient *T*. Then, a control coefficient *t* value must be selected to segment the image into targets and background.

The value of *t* has a significant effect on the detection results. A larger *t* value reduces the number of false positives and increases the number of false negatives; conversely, a smaller *t* value reduces the number of false negatives and increases the number of false positives. Therefore, the *t* value must be adjusted repeatedly to achieve the best balance between the false positive rate and the false negative rate. Usually, *t* value is a parameter set subjectively based on experience or repeated experiments. Empirical parameters have certain reference value, but there are often differences between data from different sources or data from different coverage areas, so empirical parameters are not necessarily applicable. On the other hand, the experimental method usually can only select a parameter with relatively good detection results in the limited times of adjustment process, but it is also not necessarily the optimal parameter.

#### 3.2.2. Estimation of the Optimal False Alarm Rate Control Coefficient *t* Based on Maximum Entropy

Segmenting the image of coefficient T distribution into target and background classes is a binary classification problem. In this study, the key to the problem is how to find an optimal control coefficient *t* from the set of coefficient *T* ($\left\{T_{1,}T_{2,}T_{3,\cdots }T_{n,}\right\}$) to segment it into two categories and achieve the best balance between target detection rate, false alarm rate and missed detection rate. Therefore, we use the maximum entropy index and a binary classification method to solve this problem [36,37].

Under the maximum entropy principle, to predict the probability distribution of a random event, our prediction should meet all the known conditions, and we should not make any subjective assumptions about the unknown situation. This principle produces the most uniform probability distribution and the smallest predicted risk. Because the information entropy of the probability distribution is the highest at this time, we call this model the “maximum entropy model” [38,39,40,41,42]. Here, we consider all the *T* values as a probability distribution (Figure 10). The control coefficient *t* value $T_{i}$ that maximizes the information entropy of the target and background ($E_{s}=E_{t}+E_{b}$) is the optimal segmentation threshold, that is, the optimal false alarm rate control coefficient *t*. The calculation method is shown below.

According to the principle of information entropy, the probability distribution function of the possible results of each random variable $T_{i}$ in the statistical frequency distribution of the coefficient *T* is $P\left(T_{i}\right)$, this satisfies [43] as follows:(3)$$0\leq P\left(T_{i}\right)\leq 1,\sum^{\;}P\left(T_{i}\right)=1$$

Therefore, the entropy of the coefficient T can be expressed by Equation (4) [44,45]:(4)$$E=-\overset{T_{n}}{\underset{T_{1}}{\sum }}\left(P_{i}\cdot \ln P_{i}\right)$$
where $P\left(T_{i}\right)=\frac{n}{N}$, *N* represents the number of *T* values, and *n* represents the number of $T_{i}$.

Similar to the above calculation method, when the control coefficient *t* is taken as $T_{i}$ ($T_{1}<T_{i}<T_{n}$), the set of the entire coefficient *T* is divided into two sets: background and target, the information entropy *E_t_* of the target pixels, the information entropy *E_b_* of the background pixels, and their sum *E_s_* is shown in Equations (5)–(7).
(5)$$E_{b}=-\overset{T_{i}}{\underset{T_{1}}{\sum }}\left(P\left(T_{i}\right)\cdot \ln P\left(T_{i}\right)\right)$$
where $P\left(T_{i}\right)=\frac{n_{b}}{N_{b}}$, $N_{b}$ represents the number of *T* values in the background set, and $n_{b}$ represents the number of $T_{i}$ in the background set:(6)$$E_{t}=-\overset{T_{n}}{\underset{T_{i}}{\sum }}\left(P\left(T_{i}\right)\cdot \ln P\left(T_{i}\right)\right)$$
where $P\left(T_{i}\right)=\frac{n_{t}}{N_{t}}$, $N_{t}$ represents the number of *T* values in the target set, and $n_{t}$ represents the number of $T_{i}$ in the target set:(7)$$E_{s}=E_{b}+E_{t}=-\overset{T_{i}}{\underset{T_{1}}{\sum }}\left(P\left(T_{i}\right)\cdot \ln P\left(T_{i}\right)\right)-\overset{T_{n}}{\underset{T_{i}}{\sum }}\left(P\left(T_{i}\right)\cdot \ln P\left(T_{i}\right)\right)$$

Iterate over each of the coefficients $T_{i}$ in the value range of ($T_{1}<T_{i}<T_{n}$) to get a set of the sum of the information entropy ($\left\{E_{s}\right\}$). Then, according to the maximum value of it ($Max\left\{E_{s}\right\}$), the optimal false alarm rate control coefficient t is obtained.

For example, when performing target detection with CFAR on the test area (Figure 9b), the distribution of the coefficient $T_{i}$ can be obtained (Figure 11). The sum of entropy is the highest when $T_{i}$ = 0.7 and $Max\left\{E_{s}\right\}$ = 4.6 (Figure 12). At this time, based on the real offshore platform data of the experimental area, the target detection achieves the highest true positive rate and the lowest false negative and false positive rates [10] (Figure 13). At this point, the detected targets include offshore platforms, ships, lighthouses, and other offshore artificial facilities.

### 3.3. Elimination of Ship Targets

The targets detected by the two-parameter CFAR algorithm based on maximum entropy include offshore platforms and ships; thus, it is necessary to distinguish the two and eliminate the interference from ships. Because offshore platforms are relatively static while ships exhibit motion characteristics [2], the ships can be removed by comparing the target positions of the two target results, and the position distribution of the offshore platforms can be obtained. Due to the imaging difference between the two Radarsat-2 SAR images and the impact of image registration error, the point positions of the targets in the two images may not coincide exactly. We conducted a neighborhood analysis on the detection result sets of the two phases [46,47,48] and identified the targets within a certain range threshold as offshore platforms and the targets outside the range threshold as ships.

First, the two-phase target raster data are vectorized into point position data and generated dot vector layers: *L_1_* and *L_2_*, respectively. Then, the neighborhood analysis is performed on the two possible target point layers. Based on layer *L_1_*, the distance thresholds from *L_1_* to *L_2_* are traversed, the distance is judged, and the points smaller than the threshold distance in *L_1_* are retained (Figure 14a). The setting of the distance threshold depends on the length and width of the offshore platform and the spatial resolution of the SAR image. The length and width of the offshore platform in the study are usually less than or equal to 150 m, and the spatial resolution of the Radarsat-2 SAR image used in this study is 100 m. In principle, the distance threshold is greater than or equal to the maximum size of the detected target in order to facilitate target discrimination. Therefore, the distance threshold *D* is set as 150 m.

## 4. Results

The two-parameter CFAR target detection method based on maximum entropy proposed in this paper was applied to the distribution of offshore platforms in the Pearl River Estuary Basin of the South China Sea. The extraction results of the optimal false alarm rate control coefficient in this method are compared with the extraction results of the empirical control coefficients. The extraction results are as follows.

### 4.1. Parameters Used in the Two-Parameter CFAR Target Detection Method Based on the Maximum Entropy of Offshore Platforms

The parameters used in the extraction method are shown in Table 3. In the image preprocessing part, we set the parameters for image correction, masking, and filtering. We used the source data file of the image to complete the image correction. The mask data were sourced from the *Atlas of China’s Petroliferous Basins*. The selected filtering method was sigma filtering. The window size was 3 × 3, and the filtering threshold was 8. In the target detection part, we set the detection window size and false alarm rate control coefficient for the two-parameter CFAR algorithm. Finally, the detection windows were set as follows: the target window was 225 m (3 × 3), the protection window was 525 m (7 × 7), the background window was 975 m (13 × 13), the coefficient was 0.7 (See Figure 15), and the distance threshold for the neighborhood analysis was 150 m.

### 4.2. Offshore Platform Extraction Results

The two-phase RadarSat-2 SAR images (the imaging dates were March 2014 and October 2014) were used to pre-extract the offshore platforms in the Pearl River Estuary Basin using the two-parameter CFAR method based on maximum entropy. Then, the neighborhood analysis method was used to eliminate any moving ship targets from the possible offshore platform targets extracted from the two-phase images. After the neighborhood analysis, the remaining targets are the final offshore platform targets. From the results, a total of 42 possible offshore platform targets were extracted from the Pearl River Estuary Basin images. In the results, the offshore platforms are aggregated in the western and central eastern parts of the Pearl River Estuary Basin. Among these, the number of offshore platforms located in the western part of the basin is relatively small and concentrated, while the number of offshore platforms in the central and eastern parts is relatively large and scattered (Figure 16). The spatial distribution of the offshore platforms is consistent with their true distribution.

### 4.3. Accuracy Evaluation of the Automatic Extraction Method for Offshore Platforms

In this study, we evaluated the results of offshore platform extraction using three indicators in our previous published paper: true positive (TP) rate, false negative (FN) rate and false positive (FP) rate [10].

The evaluation results are shown in Table 4, where *S* is the actual number of offshore platforms obtained by the field survey, *N_TP_* is the number of correctly extracted offshore platforms. *N_FN_* is the number of offshore platforms omitted from the extraction, and *N_FP_* is the number of offshore platforms that were extracted incorrectly. The results in Table 4 show that the two-parameter CFAR target detection method based on maximum entropy extracted 42 offshore platforms located in the Pearl River Estuary Basin of the South China Sea. Among these, 39 were correctly extracted, one was omitted and two were extracted incorrectly, achieving a TP rate of 97.5%, a FN rate of 2.5% and a FP rate of 5%. These results demonstrate that the proposed detection method has good performance for automatically extracting offshore platforms. Although certain false negative and false positive rates exist, their ratio is relatively low, which satisfies the extraction accuracy requirements of practical applications. The accuracy of the two empirical control coefficients is lower than that of the optimal control coefficient calculated based on the maximum entropy method, which also proves the superiority of the proposed method.

### 4.4. High-Resolution Image Comparison Analysis

This study used 31 high-resolution GF-1 images to compare and analyze the offshore platform extraction results in the Pearl River Estuary Basin based on the optimal false alarm control coefficient (*t = T_i_* = 0.7). Through careful visual identification, we found that among the 42 offshore platforms extracted, 39 were correctly extracted, one was omitted and two were incorrectly extracted. This result is consistent with the results obtained from the field survey. According to the study needs, we divided four concentrated offshore platform distribution areas in the Pearl River Estuary Basin (Figure 17) to compare and analyze the offshore platforms in the high-resolution images. We can use TP to express the number of correctly extracted offshore platforms, verify that an extracted offshore platform truly exists and show that the automatic extraction was correct. Omitted offshore platform is reflected by FN and represents undetected offshore platforms. Incorrectly extracted offshore platforms are reflected by FP and represent that a detected offshore platform is not a true offshore platform—it may be a ship or other maritime target (Figure 18).

The results show that 39 offshore platforms were correctly detected using the two-parameter CFAR target detection method based on maximum entropy, while two targets were incorrectly detected (target No. 19 in Figure 18b and target No. 42 in Figure 18d). The GF-1 image shows that target No. 19 is actually a ship. Because its relative position does not move between the two-phase imagery, the method mistakenly considered it to be an offshore platform. Target No. 42 is actually a lighthouse. Because the lighthouse is strongly reflected in the SAR image and has positional characteristics similar to an offshore platform, it is also mistakenly extracted as an offshore platform (Figure 18a,d). In addition, one offshore platform was omitted. In the GF-1 imagery, target No. 37 is actually an offshore platform. Through visual interpretation, the size of this offshore platform was found to be smaller than the other offshore platforms, which may be the main reason why this platform was omitted. This analysis and comparison of the high-resolution images further verify the feasibility and robustness of the two-parameter CFAR target detection method based on maximum entropy.

## 5. Discussion

### 5.1. Advantages and Limitations of the Two-Parameter CFAR Target Detection Method Based on Maximum Entropy

First of all, the common feature between this method and the traditional two-parameter CFAR method is that they use local window operation to detect the target, which has a certain adaptive ability to complex background clutter changes. In previous studies, the value of the false alarm rate control coefficient *t* in the two-parameter CFAR algorithm was determined by parameter estimation based on experimental training [49] or by setting its size according to the range of empirical values [50]. It should be noted that both are subjective parameter setting methods. As SAR image data obtained in different research areas, different periods and different imaging environment conditions are also different to some extent, empirical parameters are not necessarily applicable at this time. On the other hand, the way of setting parameters through experiments also needs to be adjusted many times to achieve a relatively good precision effect. The method of this study is to introduce the maximum entropy theory on the basis of the traditional two-parameter CFAR method window operation results and combine the two methods to give an optimal control coefficient from an objective perspective. This research experiment also proves that the optimal control coefficient calculated by this method can obtain the highest target detection accuracy, which will reduce the difficulty of target detection application of offshore platforms based on SAR data.

Of course, this method also has shortcomings. First of all, the two-parameter CFAR algorithm itself uses the method of window operation to make statistics for each pixel, which requires a large amount of computation and a long time. Secondly, the calculation of maximum entropy is also of high computational complexity. Both of these processes are time-consuming computations, which have a certain impact on computational efficiency. In future work, the computational efficiency of this method could be further improved by adopting parallel or high-performance computing techniques.

### 5.2. The Influence of Ship Targets on the Extraction Result

To address the disturbances caused by ships during offshore platform extraction, we used a two-phase image to eliminate moving ship targets by comparing them to the relatively invariant positions of the offshore platforms. The RadarSat-2 SAR data from March and October 2014 used in this experiment did not overlap the fishing moratorium period (the fishing moratorium in the South China Sea in 2014 lasted from May to August); thus a large number of ships were active in the waters of the study area. This situation may result in different ship targets being detected at a given location or the same ship being detected at a given location when it has not moved. In addition, floating production storage and offloading (FPSO) ships are sometimes present near offshore platforms, and some remain static for a short time. For example, target No. 19 is shown to be a ship in the GF-1 image (Figure 18b); this error was caused by ship activity. In subsequent studies, the interference of ship targets could be eliminated by including automatic identification system (AIS) ship data or by assessing the differences in imaging characteristics between offshore platforms and ships in medium-resolution SAR images.

### 5.3. The Interference of Offshore Artificial Targets on Offshore Platform Extraction

The above offshore platform extraction results show that the two-parameter CFAR target detection method based on maximum entropy achieves good detection performance. Nevertheless, it still results in a false positive rate of approximately 5%. In contrast, using the high-resolution images, we found that one of the offshore platforms detected by mistake was actually a lighthouse (target No. 42) (Figure 18d). The artificial targets that affect the offshore platform extraction mainly include lighthouses and reefs. In SAR images, offshore lighthouses have obvious reflective characteristics, that appear as bright pixels that are significantly different compared with background clutter; however, similar to offshore platforms, they are static and can easily be mistaken for offshore platforms. Compared with the sea surface, the radar wave reflections of reefs are obviously different, and their position is static; consequently, the radar images show obvious bright pixels. In view of the influence of offshore artificial targets on offshore platform extraction, the vector data of existing lighthouses and reefs in the study area could be collected to eliminate their influences on the results.

### 5.4. The Interference of SAR Data and Platform Size on Offshore Platform Extraction

SAR microwave imaging reflects the spatial distribution of the complex scattering echoes of the target and background in the irradiated region. From the perspective of SAR data, radar incidence angle [7], polarization mode(Ship detection by the RadarSat SAR: validation of detection model predictions) and other factors will have a significant impact on the quality of target imaging. In general, the same polarization (HH, VV) is more favorable for the detection of the target whose position is parallel to the radar line of sight, while the cross-polarization (HV, VH) is better for the detection of the target whose position is at a certain Angle to the radar line of sight [20]. If the data of the same polarization mode is adopted, it will have an impact on the detection of such targets on offshore oil and gas platforms. At the same time, it has also been shown that no matter what polarization mode is adopted, the backscattering intensity of the platform will decrease in the case of low incidence angle [7]. Therefore, the influence of incident angle should be considered before the detection of oil and gas platform, and the polarization mode should be analyzed. In addition, the size of the offshore platform will also affect the detection accuracy. Because the size, cross-sectional area and other geometric parameters of the target structure determine the reflection intensity of the radar echo. In the complex sea surface environment, the smaller platform target will generate weaker radar echo reflection intensity [51], which will affect the accuracy of target detection. For example, target 37 (Figure 18c), which was omitted, is indeed smaller than other offshore platforms in the high-resolution image. In particular, in our comparative analysis of SAR images (Figure 19), we found that the background clutter near it was relatively complex and the background changed dramatically, which may be the reason why it was not extracted.

## 6. Conclusions

Petroleum and natural gas resources are strategic in the social development of all countries. Timely and accurate acquisition of development and construction information of offshore platforms can provide a reference for all countries to formulate marine development strategies. Among the many available target extraction methods, the two-parameter CFAR algorithm can adapt to different clutter environments. In CFAR, three detection windows are slid along the image at a certain step size and used to determine whether each pixel is a target, and CFAR is the most commonly used target detection method. However, to achieve accurate target detection criteria, the false alarm rate control coefficient must be repeatedly set in this method. Moreover, the false alarm rate control coefficient is the most significant influencing factor on the detection results, which makes the detection results subjective. In this paper, the optimal threshold is estimated according to the principle of maximum entropy, and the value of *t* at the point of maximum entropy is selected as the optimal detection threshold. This approach achieves an automatic method for determining the value of the false alarm rate control coefficient. The detection result of the two-parameter CFAR target detection method based on maximum entropy is both objective and more accurate.

According to the accuracy evaluation results of the offshore platform extraction of the Pearl River Estuary Basin in China, the method extraction achieves a TP rate of 97.5%, a FN rate of 2.5% and a FP rate of 5%. This level of extraction accuracy reaches real-world needs, and its application effect is good. Our approach eliminates the need to tediously set the threshold artificially as in previous studies while also removing subjectivity from the detection results.

Although the proposed method achieved an ideal extraction effect, it still has some shortcomings. In the future, further research will be conducted to improve the anti-noise capability of the maximum entropy approach, to eliminate interference targets such as ships and to improve the computational efficiency while ensuring the correct extraction rate.

## Figures and Tables

**Figure 1 entropy-21-00556-f001:**
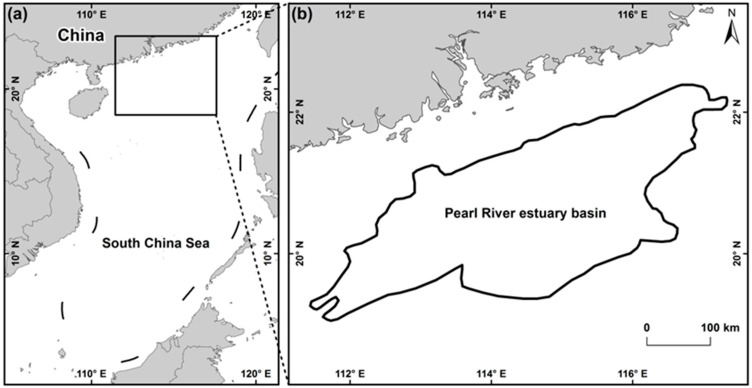
Schematic diagram of the location of the study area.

**Figure 2 entropy-21-00556-f002:**
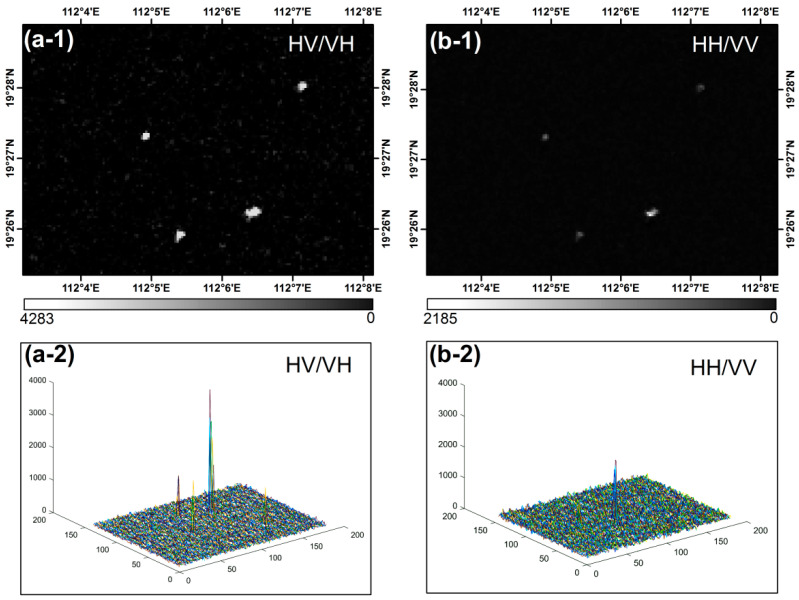
Target spectral analysis of different polarization modes. (**a-1**) and (**a-2**) are SAR images of cross-polarization mode (HV/VH) and three-dimensional images of their gray distribution, respectively. (**b-1**) and (**b-2**) are SAR images of the same polarization mode (HH/VV) and their gray distribution three-dimensional images respectively.

**Figure 3 entropy-21-00556-f003:**
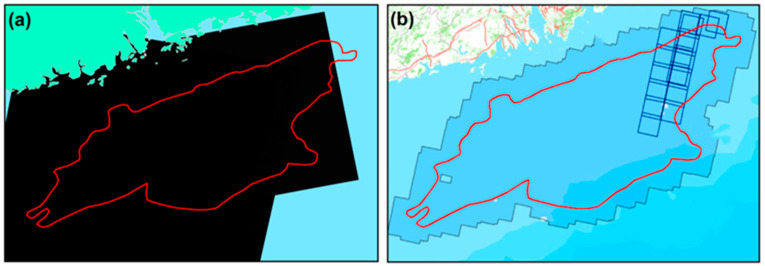
Image coverage of the Pearl River Estuary Basin: (**a**) coverage of the RadarSat-2 images in the Pearl River Estuary Basin; (**b**) coverage of the GF-1 images in the Pearl River Estuary Basin.

**Figure 4 entropy-21-00556-f004:**
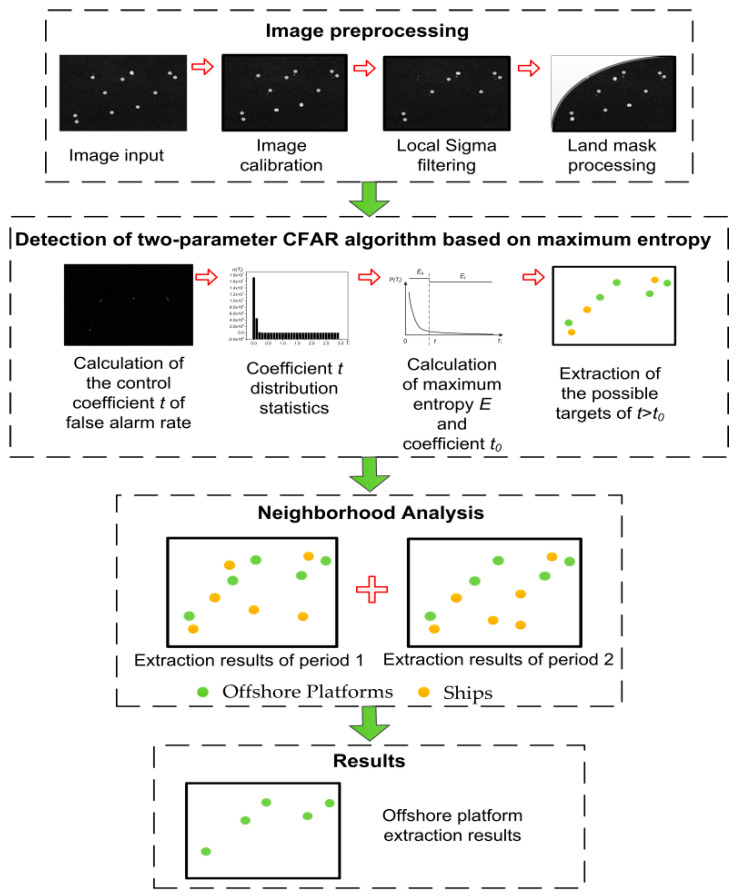
Flow chart of offshore platform detection using the two-parameter CFAR algorithm based on maximum entropy.

**Figure 5 entropy-21-00556-f005:**
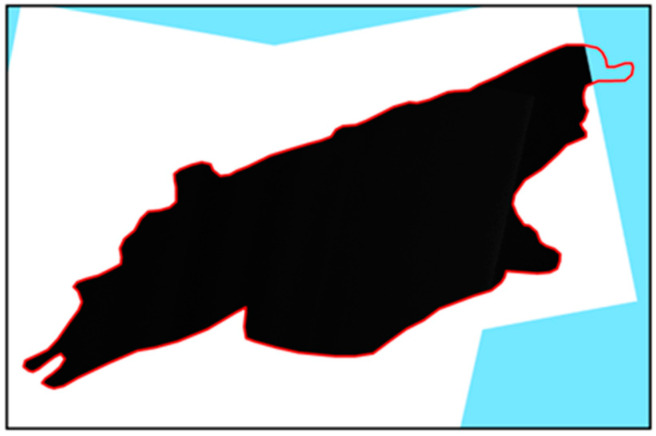
RadarSat-2 image-mask processing result.

**Figure 6 entropy-21-00556-f006:**
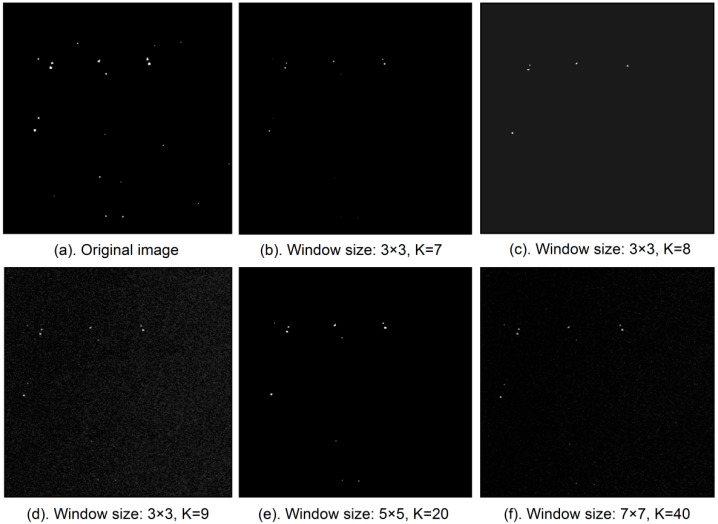
Sigma filtering parameters comparison diagram: (**a**) the original image; (**b**) the filtering result when the filtering window is 3 × 3 and K = 7; (**c**) the filtering result when the filtering window is 3 × 3 and K = 8; (**d**) the filtering result when the filtering window is 3 × 3 and K = 9; (**e**) the filtering result when the filtering window is 5 × 5 and K = 20; (**f**) the filtering result when the filtering window is 7 × 7 and K = 40.

**Figure 7 entropy-21-00556-f007:**
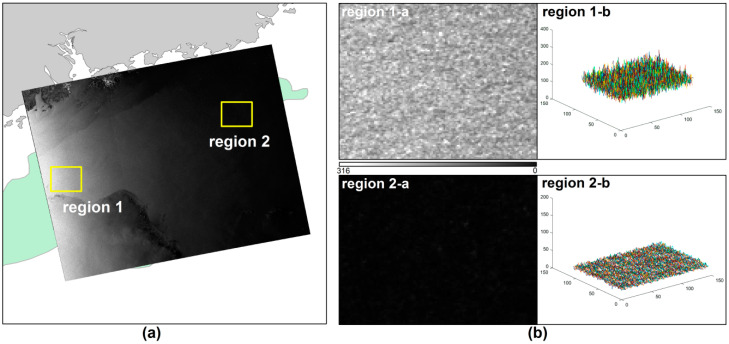
Non-uniform SAR imagery: (**a**) the coverage map of RadarSat-2 SAR in the Pearl River Estuary Basin area; (**b**) imaging features of two different regions.

**Figure 8 entropy-21-00556-f008:**
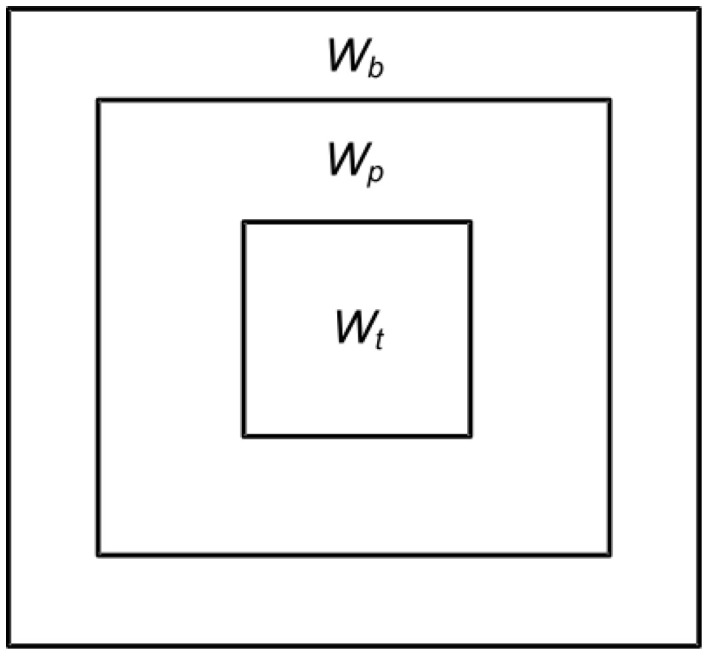
Schematic diagram of a two-parameter CFAR detection window: $W_{b}$ is the background window; $W_{p}$ is the protection window, and $W_{t}$ is the target window.

**Figure 9 entropy-21-00556-f009:**
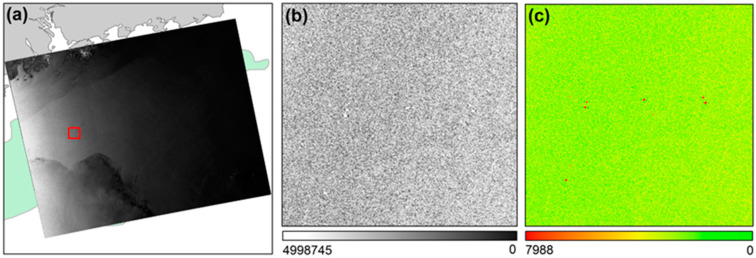
SAR image calculation process in the target window of the two-parameter CFAR algorithm: (**a**) the coverage map of the original RadarSat-2 SAR image in the Pearl River Estuary Basin: the red box is the target window; (**b**) the image gradation distribution in the target window; (**c**) the distribution of the coefficient *T*.

**Figure 10 entropy-21-00556-f010:**
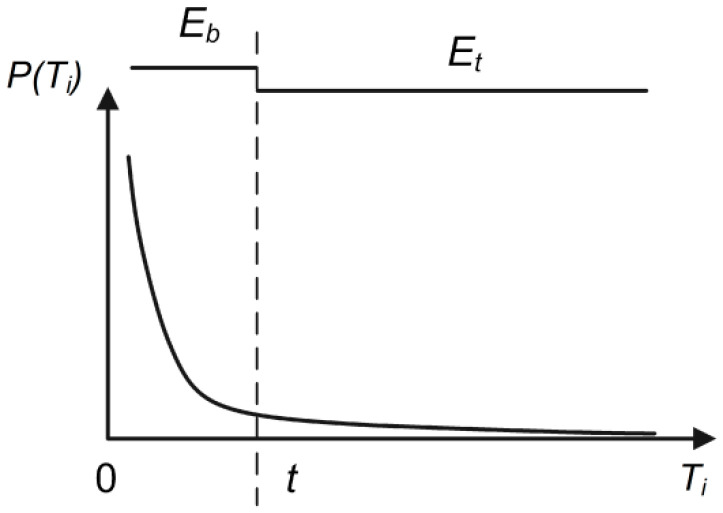
Schematic diagram of optimal false alarm rate control coefficient *t* estimation based on maximum entropy. *E_b_* represents the information entropy of background pixels, and *E_t_* represents the information entropy of target pixels.

**Figure 11 entropy-21-00556-f011:**
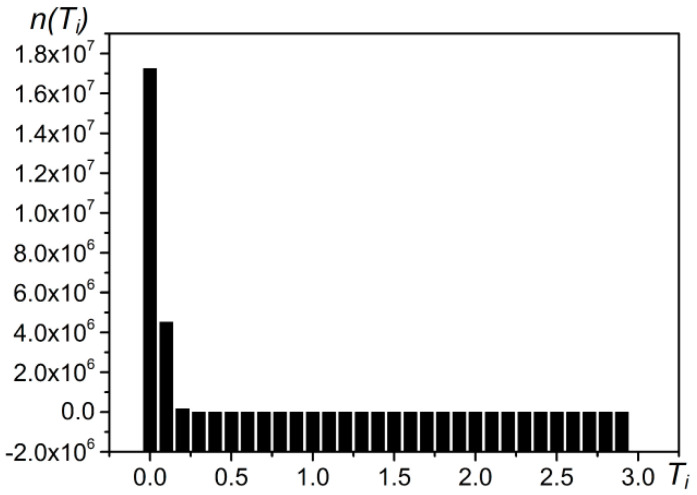
Statistical frequency graph of the coefficient *T* image in a test area. The abscissa represents the *T* value and the ordinate represents the frequency.

**Figure 12 entropy-21-00556-f012:**
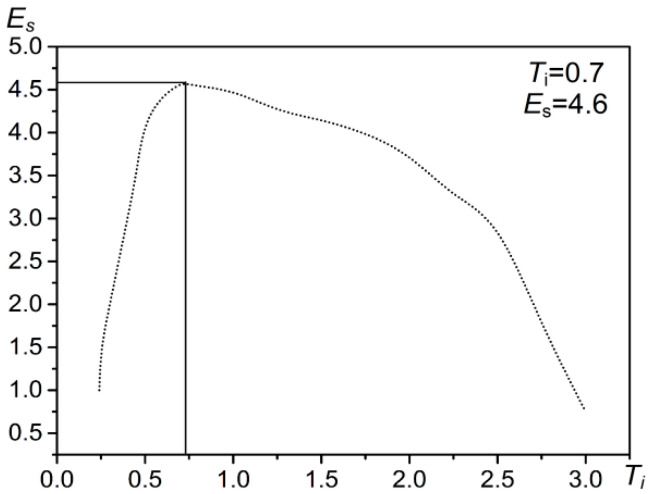
Maximum entropy operation result of coefficient $T_{i}$

**Figure 13 entropy-21-00556-f013:**
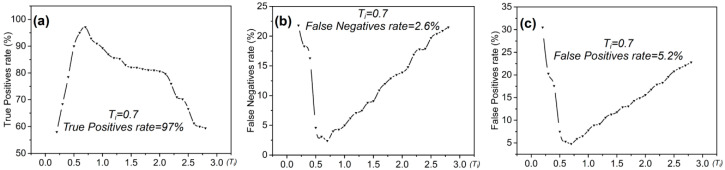
Target detection accuracy distribution diagram of the two-parameter CFAR algorithm based on maximum entropy: (**a**) the true positive distribution map; (**b**) the distribution map of the false negative rate; and (**c**) the distribution diagram of the false positive rate.

**Figure 14 entropy-21-00556-f014:**
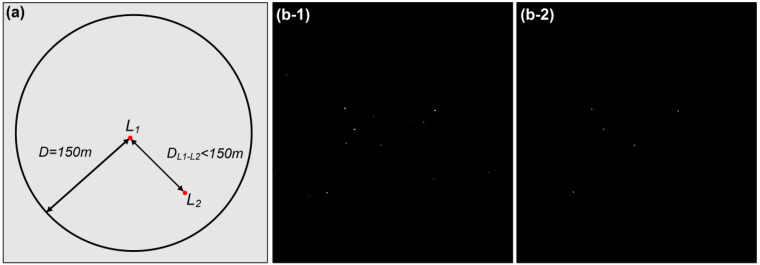
Schematic diagram of the principle and results of neighborhood analysis: (**a**) a schematic diagram of the principle of neighborhood analysis. The target point in the figure represents the same target point in layers *L_1_* and *L_2_*; (**b-1**) the detection result map of a certain area in the Pearl River Estuary Basin, including targets such as offshore platforms and ships; (**b-2**) the target distribution map of the offshore platform after neighborhood analysis.

**Figure 15 entropy-21-00556-f015:**
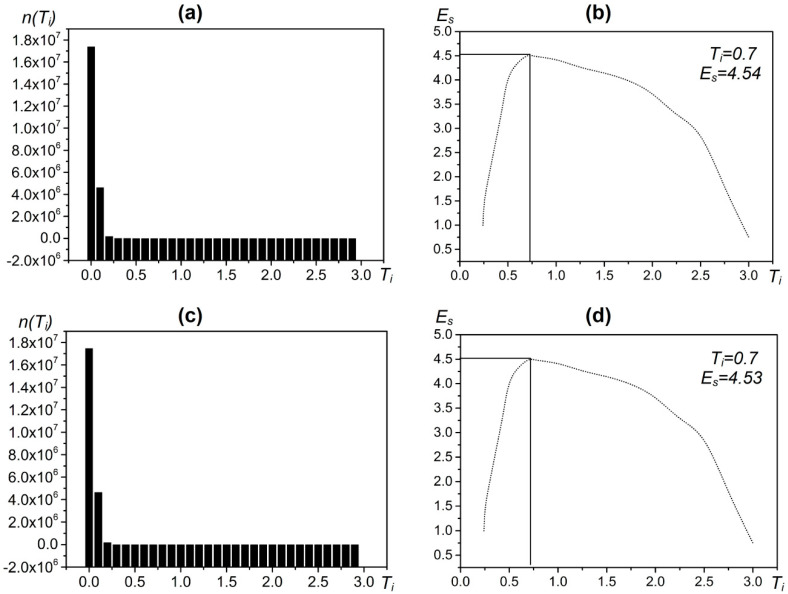
The maximum entropy calculation result of the coefficient $T_{i}$ of the SAR image: (**a**) the frequency distribution diagram of the coefficient $T_{i}$ of the SAR image in March 2014; (**b**) the maximum entropy result diagram of the coefficient $T_{i}$ of the SAR image in March 2014; (**c**) the frequency distribution diagram of the coefficient $T_{i}$ of the SAR image in October 2014; and (**d**) the maximum entropy result diagram of the coefficient $T_{i}$ of the SAR image in October 2014.

**Figure 16 entropy-21-00556-f016:**
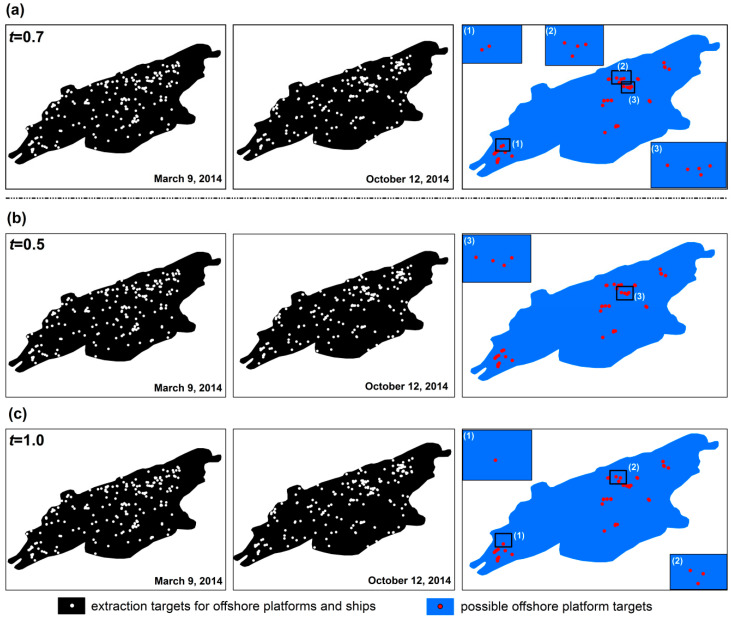
Offshore platform extraction results using a two-parameter CFAR target detection method based on maximum entropy: (**a**) is the extraction result when *t* = 0.7, (**b**) is the extraction result when *t* = 0.5, and (**c**) is the extraction result when *t* = 1.0.

**Figure 17 entropy-21-00556-f017:**
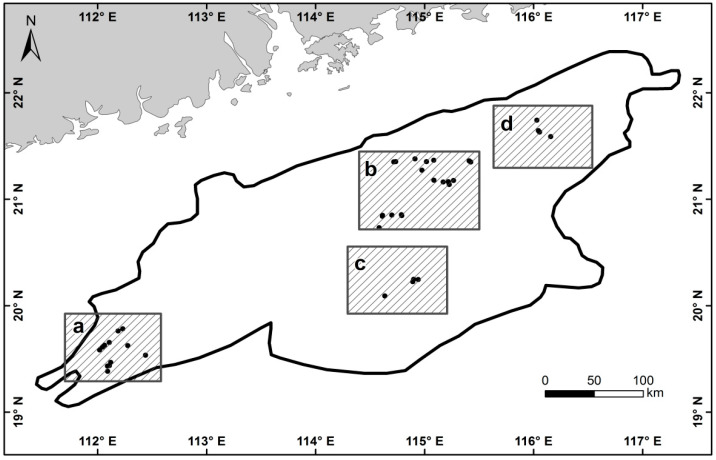
Schematic diagram of the region divisions of the high-resolution images used for comparison and analysis of the automatic extraction results: (**a**)–(**d**) represent four concentrated offshore platform distribution areas used to compare and analyze the extraction results.

**Figure 18 entropy-21-00556-f018:**
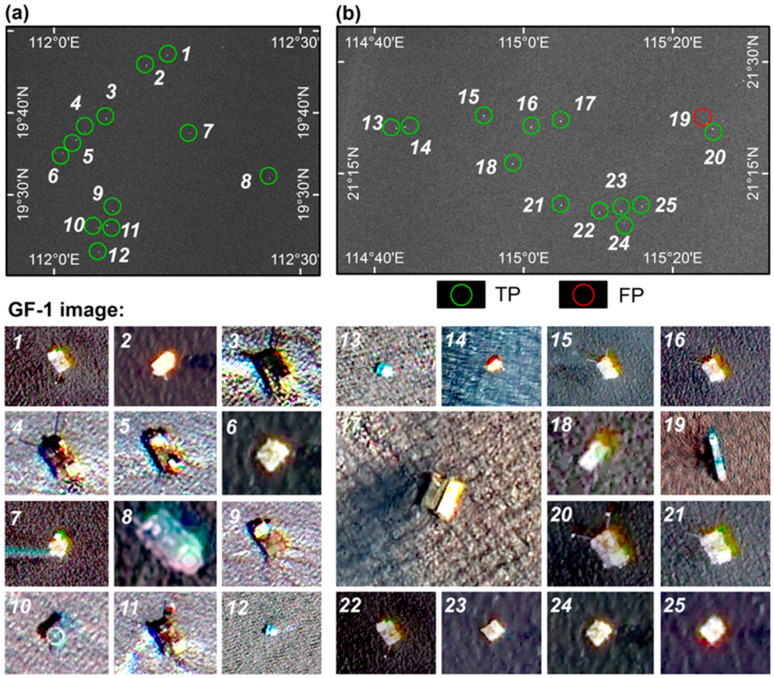
Comparison and analysis of extraction results based on high-resolution images. where (**a**), (**b**), (**c**) and (**d**) are the extraction results of the four regions of the Pearl River Estuary Basin indicated in Figure 17 and the comparison results in the high-resolution images.

**Figure 19 entropy-21-00556-f019:**
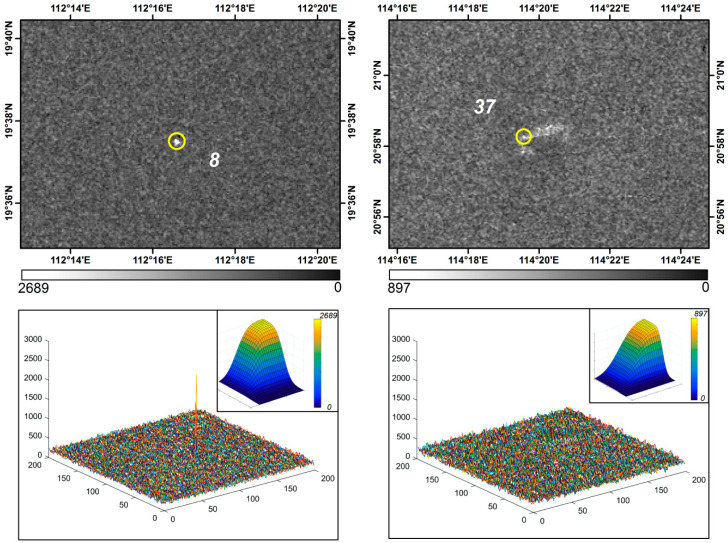
Comparative analysis of SAR images: three-dimensional intensity and background clutter model within the target detection window. Note that the sea clutter near target No. 37 is more complicated and that target No. 8 is less affected by the sea clutter.

**Table 1 entropy-21-00556-t001:** RadarSat-2 remote sensing image data. (SAR = synthetic aperture radar).

No.	Satellite	Sensor	Image Mode	Polarization Mode	Spatial Resolution(/m)	Pixel Spacing (/m)	Date
1	RadarSat-2	SAR	ScanSAR Wide	VH	100 m	100 m	2014-03-09
2	RadarSat-2	SAR	ScanSAR Wide	VH	100 m	100 m	2014-10-12

**Table 2 entropy-21-00556-t002:** GF-1 remote sensing image data. (PMS = particular matter sensor).

No	Satellite	Sensor	Spatial Resolution(/m)	Time	Cloud Cover (%)
1	GF-1	PMS	2	2014-09-12	5
2	GF-1	PMS	2	2014-09-12	7
3	GF-1	PMS	2	2014-09-12	6
4	GF-1	PMS	2	2014-10-09	3
5	GF-1	PMS	2	2014-10-09	3
6	GF-1	PMS	2	2014-11-02	0
7	GF-1	PMS	2	2014-11-15	0
8	GF-1	PMS	2	2014-11-23	9
9	GF-1	PMS	2	2014-11-23	8
10	GF-1	PMS	2	2014-11-25	11
11	GF-1	PMS	2	2014-11-25	11
12	GF-1	PMS	2	2014-12-11	13
13	GF-1	PMS	2	2014-12-11	0
14	GF-1	PMS	2	2014-12-19	0
15	GF-1	PMS	2	2014-12-19	9
16	GF-1	PMS	2	2014-12-26	15
17	GF-1	PMS	2	2014-12-26	12
18	GF-1	PMS	2	2014-12-27	3
19	GF-1	PMS	2	2014-12-27	1
20	GF-1	PMS	2	2014-12-27	0
21	GF-1	PMS	2	2014-12-27	7
22	GF-1	PMS	2	2014-12-28	9
23	GF-1	PMS	2	2014-12-28	13
24	GF-1	PMS	2	2014-12-28	8
25	GF-1	PMS	2	2014-12-28	5
26	GF-1	PMS	2	2014-12-28	11
27	GF-1	PMS	2	2014-12-28	0
28	GF-1	PMS	2	2014-12-29	0
29	GF-1	PMS	2	2014-12-29	5
30	GF-1	PMS	2	2014-12-29	1
31	GF-1	PMS	2	2014-12-29	1

**Table 3 entropy-21-00556-t003:** The parameter list of the two-parameter CFAR target detection method based on maximum entropy.

Categories	Parameters
Image preprocessing	Calibration: backscattering coefficient and sensor calibration data are acquired from the metadata file
	Land mask: data of the extent of petroliferous basins provided by the “Atlas of China’s Petroliferous Basins”
	Sigma filtering: filter size: 3 × 3 (σ = 8)
Target detection	Detection thresholds of the maximum entropy based two-parameter CFAR: detection window sizes: 225 m (3 × 3), 525 m (7 × 7), 975 m (13 × 13); false alarm rate control coefficient: *t* = $T_{i}$ = 0.7. (Figure 15) Neighborhood analysis: distance threshold = 150 m.

**Table 4 entropy-21-00556-t004:** Accuracy evaluation results of the automatic extraction method for offshore platforms.

	Two-Parameter CFAR Based on Maximum Entropy	Two-Parameter CFAR	
	Optimal Control Coefficient *t* = 0.7	Empirical Control Coefficient *t* = 0.5	Empirical Control Coefficient *t* = 1
Actual number(*S*)	40	40	40
Total extraction(*N*)	42	41	40
*N_TP_*	39	38	37
*N_FP_*	1	3	1
*N_FN_*	2	0	2
*TP*	97.5%	95%	92.5%
*FP*	2.5%	7.5%	2.5%
*FN*	5%	0	5%

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
