# Peer review of "Offshore Platform Extraction Using RadarSat-2 SAR Imagery: A Two-Parameter CFAR Method Based on Maximum Entropy"

_entropy, 2019, doi:10.3390/e21060556_

Round 1

Reviewer 1 Report

General comments:

The manuscript presents a novel methodology to detect off shore platforms using SAR data. Although the detection of off-shore platforms is not new, the novelty in the methodology (maximising the entropy) is enough to make the manuscript novel and worth publishing. There are however some major point related to stating the background literature and explaining better the details of the methodology that at the moment are still very general.    

Major points:

Comparison: This is an interesting methodology, however it is important that proposed algorithms are compared to state-of-the-art. One important candidate for comparison is the CFAR where t is set to a standard value, say 6 (or any other that provides good detection). Since the entropy maximisation is time consuming it is important to evaluate the advantage with respect of using simpler methodologies. 

Page 1 line 27: The sentence “… obtains the best target detection performance” cannot be stated unless a comparison with other detectors is performed. Please address the previous point and then you can write this.

References: The literature review should be improved. There are at least two articles dealing with detecting offshore platforms that should be include. Please add:

A. Marino, D. Velotto, F. Nunziata, “Offshore Metallic Platforms Observation Using Dual-Polarimetric TS-X/TD-X Satellite Imagery: A Case Study in the Gulf of Mexico”, IEEE Journal of Selected Topics in Applied Earth Observations and Remote Sensing, 10(10), Oct. 2017.

Q. Xing, Meng R., Lou M.J., Bing L., Liu X., “Remote Sensing of Ships and Offshore Oil Platforms and Mapping the Marine Oil Spill Risk Source in the Bohai Sea” Aquatic Procedia, 3, 2015, 127-132.

The second also consider time series of images as you. You work has the novelty of using the maximum entropy and therefore it deserves to be published.

Another thing that is missing is a better coverage of more advanced methodologies that use polarimetry of spectral analysis (as for the first paper). Please add at least a mention that if we have polarimetry or if we perform spectral analysis we can obtain better results, although this impact on computational time and storage space. Some articles that are relevant to this study are:

Polarimetry:

G. D. De Grandi, J.-S. Lee, D. L. Schuler, Target detection and texture segmentation in polarimetric SAR images using a wavelet frame: Theoretical aspects, IEEE Transactions on Geoscience and Remote Sensing 45 (11) 2007.

Marino, A. “A notch filter for ship detection with polarimetric SAR data.” IEEE Journal

of Selected Topics Applied Earth Observations and Remote Sensing 6 (3): 1219-1232, 2013.

R. Touzi, J. Hurley, P.W. Vachon, C. Avenue, Optimization of the degree of polarization for enhanced ship detection using polarimetric RADARSAT-2, IEEE Transactions on Geoscience and Remote Sensing 53 (10), 2015

Sub aperture:

A. Arnaud, “Ship detection by SAR interferometry,” in Geoscience and Remote Sensing Symposium, 1999. IGARSS’99 Proceedings. IEEE 1999 International, vol. 5. IEEE, 1999, pp. 2616–2618.

A. Marino, M. J. Sanjuan-Ferrer, I. Hajnsek, and K. Ouchi, “Ship detection with spectral analysis of synthetic aperture radar: a comparison of new and well-known algorithms,” Remote Sensing, vol. 7, no. 5, pp. 5416–5439, 2015.

C. Brekke, S. N. Anfinsen, and Y. Larsen, “Subband extraction strategies in ship detection with the subaperture cross-correlation magnitude,” IEEE Geoscience and Remote Sensing Letters, vol. 10, no. 4, pp. 786–790, 2013.

Equation 1: In CFAR detection the value of t is fixed and the threshold T calculated with equation 1. It is the final threshold that is adjusted locally depending on mean and standard deviation, not the value t. I am not sure how you are changing the value of t. What is the dependency of t on sigma and mu? Please express the mathematical equation used to evaluate t, since equation 10 only expresses the value of the final threshold T, such that it is bigger than that. 

Entropy optimisation: please provide more details on the optimisation. What are the variables on which the optimisation is done and what numerical methodology you use for this?

Figure 11: There is something quite curious in this figure. The entropy has a local minimum between the value of t_0 included in [0.3, 0.7]. This means that there is some threshold t_0 that if you reduce it you will have better performance and if you increase it you will get again better performance. Or in general a more uniform distribution. Since pixel intensity is a 1 dimensional feature space, this appears to me to be impossible, because the amount of pixels above a threshold t_1 cannot be more if you increase the threshold to t_2. And the same is for number of pixels below a threshold when you reduce it. Please check that there is no errors in this calculations and if not please explain how a 1D variable can have a leap or a warp so that in an interval, higher numbers actually appear smaller.

Page 11 line 33: If I understood well, you apply an initial 3x3 sigma filter, then you apply again a 3x3 filter on the obtained image. You are double filtering. Is this really convenient? You are not really improving much statistics by doing that? Have you tried just to use the single pixel, after the sigma filter, as the test area? And still apply the same training area.

Minor:

Page 5 line 160: what do you mean with “two-phase”? Do you mean complex number?

Page 6 line 177: please change “radiation” into “radiometric”.

Page  7 line 208: Please change “non-uniform” into “noisy”.

Figure 13: Your search in the local area to evaluate if a target is a platform is a good thing. Misplacement is mostly due to errors in corregistration. Please add this in the manuscript.

Author Response

Dear professor,

Thank you very much for your careful review and sincere suggestions. We have revised the article according to your comments, the details are as follows:

Comment 1:

Comparison: This is an interesting methodology, however it is important that proposed algorithms are compared to state-of-the-art. One important candidate for comparison is the CFAR where t is set to a standard value, say 6 (or any other that provides good detection). Since the entropy maximization is time consuming it is important to evaluate the advantage with respect of using simpler methodologies.

Page 1 line 27: The sentence “… obtains the best target detection performance” cannot be stated unless a comparison with other detectors is performed. Please address the previous point and then you can write this.

Response 1: We carefully considered the suggestion given by the reviewer and compared the method proposed in this paper with the traditional two-parameter CFAR algorithm in the experimental application part. We first selected two empirical coefficients for target detection, then apply the optimal control coefficient calculated by the method in this paper to target detection, and finally compared the detection results and precision. The results show that the detection accuracy of the proposed method is better than that of the traditional method. The accuracy comparison and conclusion of the experimental part are as follows.

Table 4. Accuracy evaluation results of the automatic extraction method for offshore platforms.

Two-parameter CFAR based on maximum entropy

Two-parameter CFAR

Optimal control coefficient t=0.7

empirical control coefficient

t=0.5

empirical control coefficient

t=1

Actual numberS

40

40

40

Total extractionN

42

41

40

NTP

39

38

37

NFP

1

3

1

NFN

2

0

2

TP

97.5%

95%

92.5%

FP

2.5%

7.5%

2.5%

FN

5%

0

5%

The evaluation results in Table 4 show that the two-parameter CFAR target detection method based on maximum entropy extracted 42 offshore platforms located in the Pearl River Estuary Basin of the South China Sea. Among these, 39 were correctly extracted, one was omitted and two were extracted incorrectly, achieving a TP rate of 97.5%, a FN rate of 2.5% and a FP rate of 5%. These results demonstrate that the proposed detection method has good performance for automatically extracting offshore platforms. Although certain false negative and false positive rates exist, their ratio is relatively low, which satisfies the extraction accuracy requirements of practical applications. The accuracy of the two empirical control coefficients is lower than that of the optimal control coefficient calculated based on the maximum entropy method, which also proves the superiority of the proposed method.

-----------------------------------------------------------------------------------------------------

Comment 2:

References: The literature review should be improved. There are at least two articles dealing with detecting offshore platforms that should be include. Please add:

A. Marino, D. Velotto, F. Nunziata, “Offshore Metallic Platforms Observation Using Dual-Polarimetric TS-X/TD-X Satellite Imagery: A Case Study in the Gulf of Mexico”, IEEE Journal of Selected Topics in Applied Earth Observations and Remote Sensing, 10(10), Oct. 2017.

Q. Xing, Meng R., Lou M.J., Bing L., Liu X., “Remote Sensing of Ships and Offshore Oil Platforms and Mapping the Marine Oil Spill Risk Source in the Bohai Sea” Aquatic Procedia, 3, 2015, 127-132.

The second also consider time series of images as you. You work has the novelty of using the maximum entropy and therefore it deserves to be published.

Another thing that is missing is a better coverage of more advanced methodologies that use polarimetry of spectral analysis (as for the first paper). Please add at least a mention that if we have polarimetry or if we perform spectral analysis we can obtain better results, although this impact on computational time and storage space. Some articles that are relevant to this study are:

Polarimetry:

G. D. De Grandi, J.-S. Lee, D. L. Schuler, Target detection and texture segmentation in polarimetric SAR images using a wavelet frame: Theoretical aspects, IEEE Transactions on Geoscience and Remote Sensing 45 (11) 2007.

Marino, A. “A notch filter for ship detection with polarimetric SAR data.” IEEE Journal of Selected Topics Applied Earth Observations and Remote Sensing 6 (3): 1219-1232, 2013.

R. Touzi, J. Hurley, P.W. Vachon, C. Avenue, Optimization of the degree of polarization for enhanced ship detection using polarimetric RADARSAT-2, IEEE Transactions on Geoscience and Remote Sensing 53 (10), 2015

Sub aperture:

A. Arnaud, “Ship detection by SAR interferometry,” in Geoscience and Remote Sensing Symposium, 1999. IGARSS’99 Proceedings. IEEE 1999 International, vol. 5. IEEE, 1999, pp. 2616–2618.

A. Marino, M. J. Sanjuan-Ferrer, I. Hajnsek, and K. Ouchi, “Ship detection with spectral analysis of synthetic aperture radar: a comparison of new and well-known algorithms,” Remote Sensing, vol. 7, no. 5, pp. 5416–5439, 2015.

C. Brekke, S. N. Anfinsen, and Y. Larsen, “Subband extraction strategies in ship detection with the subaperture cross-correlation magnitude,” IEEE Geoscience and Remote Sensing Letters, vol. 10, no. 4, pp. 786–790, 2013.

Response 2: Thanks for the reviewer’s good suggestion, and we all agree with the reviewer’s comments. Firstly, we made a revision to the introduction of this paper and quoted the relevant literatures. Secondly, in the data selection, discussion and other parts, descriptions and related diagrams of the influence of SAR image polarization mode, incident angle and other factors on offshore target detection are added. This makes our study more perfect and rigorous. Some of the subsections are revised as follows.

2.2 Datasets

The RadarSat-2 SAR georeferenced fine resolution (SGF, SAR Georeferenced Fine Resolution) images (Table 1, Fig 2-a) of the 2-width scan (ScanSAR Wide) mode was collected for offshore platform extraction. This image product underwent processing such as ground distance conversion, multi-view processing, and image calibration. The dates of the two images are 2014-03-09 and 2014-10-12, and the time interval is 7 months, which is beneficial to target detection during the two phases of the neighborhood analysis to eliminate the moving ship target. The images have a spatial resolution of 100 m, VH polarization and a width of 500 km × 500 km. In general, the co-polarization (HH, VV) mode is more favorable for a target whose structure is parallel to the radar’s viewing direction, and cross polarization (HV, VH) better detects targets at a certain angle from the radar’s viewing angle [1]. The deck heights of main offshore platform structures are generally located approximately 20 m above the water surface and form a certain angle with the radar viewing direction. Therefore, the selection of VH-polarized data is more conducive to target detection(Fig.2).

Figure 2. Target spectral analysis of different polarization modes

5.4. The interference of SAR data and platform size on offshore platform extraction

SAR microwave imaging reflects the spatial distribution of the complex scattering echoes of the target and background in the irradiated region. From the perspective of SAR data, radar incidence angle[7], polarization mode(Ship detection by the Radarsat SAR: validation of detection model predictions) and other factors will have a significant impact on the quality of target imaging. In general, the same polarization (HH, VV) is more favorable for the detection of the target whose position is parallel to the radar line of sight, while the cross-polarization (HV, VH) is better for the detection of the target whose position is at a certain Angle to the radar line of sight [20]. If the data of the same polarization mode is adopted, it will have an impact on the detection of such targets on offshore oil and gas platforms. At the same time, it has also been shown that no matter what polarization mode is adopted, the backscattering intensity of the platform will decrease in the case of low incidence angle[7]. Therefore, the influence of incident angle should be considered before the detection of oil and gas platform, and the polarization mode should be analyzed. In addition, the size of the offshore platform will also affect the detection accuracy. Because the size, cross-sectional area and other geometric parameters of the target structure determine the reflection intensity of the radar echo. In the complex sea surface environment, the smaller platform target will generate weaker radar echo reflection intensity[51], which will affect the accuracy of target detection. For example, target 37 (Fig. 18-c), which was omitted, is indeed smaller than other offshore platforms in the high-resolution image. In particular, in our comparative analysis of SAR images (Fig. 19), we found that the background clutter near it was relatively complex and the background changed dramatically, which may be the reason why it was not extracted.

Figure 19. Comparative analysis of SAR images: three-dimensional intensity and background clutter model within the target detection window. Note that the sea clutter near target No. 37 is more complicated and that target No. 8 is less affected by the sea clutter.

-----------------------------------------------------------------------------------------------------

Comment 3:

Equation 1: In CFAR detection the value of t is fixed and the threshold T calculated with equation 1. It is the final threshold that is adjusted locally depending on mean and standard deviation, not the value t. I am not sure how you are changing the value of t. What is the dependency of t on sigma and mu? Please express the mathematical equation used to evaluate t, since equation 1 only expresses the value of the final threshold T, such that it is bigger than that.

Response 3: From other people's point of view, our description of this part is really not clear and appropriate enough. We have reorganized this part, including the definition of the formula and the process of SAR image from intensity value to coefficient value. The revised content is shown below.

The two-parameter CFAR detection algorithm needs to create three detection windows centered on each pixel during target detection: a background window , a protection window  and a target window  (shown in Fig. 8). The background window represents the statistical information of the sea clutter background. The protection window helps ensure that the target pixel does not leak into the background window. The target window is mainly used to detect offshore platforms and ship targets. Generally, the size of the target window is set to the size of the largest target in the detected image; the size of the protection window is twice that of the target window; and the background window is set to the size of the protection window + 2n, where the value of n is usually 3 [2]. Because the directions of offshore platforms and ship targets are not fixed, the detection window is a square. The two-parameter CFAR algorithm performs target detection by calculating the mean  of all the pixels in the background window and the background standard deviation . If the grey value of a pixel in the target window is , the criterion for determining whether a pixel in the target window is a target is [3]:

.                                      (1)

When , the pixel is a considered target. Otherwise, it is considered to be background. Here, t is a standardized factor and is usually a constant (also known as the constant false alarm rate control coefficient) that controls the false alarm rate [4, 5].

Here, we transform formula (1) to obtain a function() of the grayscale value() of pixels in the target window, and the mean() and standard deviation() of the background window. The result of this function can be understood as a set of coefficients T. In fact, the target detection process is to determine whether the pixel is a target by comparing the numerical value between the coefficient T and the constant false alarm rate control coefficient t, as shown in formula (2).

                          (2)

Applying Formula (2) to the SAR image in Fig. 8-b obtains the distribution of the coefficient T with respect to the image gradation (Fig. 9-c). Because of the window calculation with local adaptive ability, there is obvious visual separability between the target and the background in the distribution image of coefficient T. Then, a control coefficient t value must be selected to segment the image into targets and background.

Figure 9. SAR image calculation process in the target window of the two-parameter CFAR algorithm: (a) the coverage map of the original RadarSat-2 SAR image in the Pearl River Estuary Basin: the red box is the target window; (b) the image gradation distribution in the target window; (c) the distribution of the coefficient T.

The value of t has a significant effect on the detection results. A larger t value reduces the number of false positives and increases the number of false negatives; conversely, a smaller t value reduces the number of false negatives and increases the number of false positives. Therefore, the t value must be adjusted repeatedly to achieve the best balance between the false positive rate and the false negative rate. Usually, t value is a parameter set subjectively based on experience or repeated experiments. Empirical parameters have certain reference value, but there are often differences between data from different sources or data from different coverage areas, so empirical parameters are not necessarily applicable. On the other hand, the experimental method usually can only select a parameter with relatively good detection results in the limited times of adjustment process, but it is also not necessarily the optimal parameter.

-----------------------------------------------------------------------------------------------------

Comment 4:

Entropy optimisation: please provide more details on the optimisation. What are the variables on which the optimisation is done and what numerical methodology you use for this?

Response 4: The problem of this part is that we do not describe it in detail and properly. The method of this part is to use the maximum entropy theory to calculate an optimal coefficient from the aforementioned conversion result (coefficient Ti) as the constant false alarm rate control coefficient, so that the target and the background can be reasonably distinguished. The final output result of this method is a coefficient Ti, whose advantage is that it is an objective parameter and can make the classification accuracy the highest. We have made significant revisions to the definition of this section, as shown below.

3.2.2 Estimation of the optimal false alarm rate control coefficient t based on maximum entropy

Segmenting the image of coefficient T distribution into target and background classes is a binary classification problem. In this study, the key to the problem is how to find an optimal control coefficient t from the set of coefficient T ()to segment it into two categories and achieve the best balance between target detection rate, false alarm rate and missed detection rate. Therefore, we use the maximum entropy index and a binary classification method to solve this problem [6, 7].

Under the maximum entropy principle, to predict the probability distribution of a random event, our prediction should meet all the known conditions, and we should not make any subjective assumptions about the unknown situation. This principle produces the most uniform probability distribution and the smallest predicted risk. Because the information entropy of the probability distribution is the highest at this time, we call this model the "maximum entropy model" [8-12]. Here, we consider all the T values as a probability distribution(Fig. 10). The control coefficient t value  that maximizes the information entropy of the target and background () is the optimal segmentation threshold, that is, the optimal false alarm rate control coefficient t. The calculation method is shown below.

Figure 10. Schematic diagram of optimal false alarm rate control coefficient t estimation based on maximum entropy. Eb represents the information entropy of background pixels, and Et represents the information entropy of target pixels.

According to the principle of information entropy, the probability distribution function of the possible results of each random variable  in the statistical frequency distribution of the coefficient T is , this satisfies [13] as follows:

.                         (3)

Therefore, the entropy of the coefficient T can be expressed by Formula (4) [14, 15]:

,                                 (4)

where , N represents the number of T values, and n represents the number of .

Similar to the above calculation method, when the control coefficient t is taken as (), the set of the entire coefficient T is divided into two sets: background and target, the information entropy Et of the target pixels, the information entropy Eb of the background pixels, and their sum Es is shown in formula (5) - (7).

                          (5)

where ,  represents the number of T values in the background set, and  represents the number of  in the background set.

                          (6)

where ,  represents the number of T values in the target set, and  represents the number of  in the target set.

        (7)

Iterate over each of the coefficients  in the value range of () to get a set of the sum of the information entropy(). Then, according to the maximum value of it (), the optimal false alarm rate control coefficient t is obtained.

For example, when performing target detection with CFAR on the test area (Fig. 9-b), the distribution of the coefficient  can be obtained (Fig. 11). The sum of entropy is the highest when =0.7 and = 4.6 (Fig. 12). At this time, the target detection achieves the highest[10] (Fig. 13). At this point, the detected targets include offshore platforms, ships, lighthouses, and other offshore artificial facilities.

Figure 11. Statistical frequency graph of the coefficient T image in a test area. The abscissa represents the T value and the ordinate represents the frequency.

Figure 12. Maximum entropy operation result of coefficient .

Figure 13. Target detection accuracy distribution diagram of the two-parameter CFAR algorithm based on maximum entropy: (a) the true positive distribution map; (b) the distribution map of the false negative rate; and (c) the distribution diagram of the false positive rate.

-----------------------------------------------------------------------------------------------------

Comment 5:

Figure 11: There is something quite curious in this figure. The entropy has a local minimum between the value of t_0 included in [0.3, 0.7]. This means that there is some threshold t_0 that if you reduce it you will have better performance and if you increase it you will get again better performance. Or in general a more uniform distribution. Since pixel intensity is a 1 dimensional feature space, this appears to me to be impossible, because the amount of pixels above a threshold t_1 cannot be more if you increase the threshold to t_2. And the same is for number of pixels below a threshold when you reduce it. Please check that there is no errors in this calculations and if not please explain how a 1D variable can have a leap or a warp so that in an interval, higher numbers actually appear smaller.

Response 5: Thank you very much for your suggestion. We have indeed ignored the problems in this figure. This is actually an improper alignment that we chose for the sake of aesthetics. We have corrected this.

-----------------------------------------------------------------------------------------------------

Comment 6:

Page 11 line 33: If I understood well, you apply an initial 3x3 sigma filter, then you apply again a 3x3 filter on the obtained image. You are double filtering. Is this really convenient? You are not really improving much statistics by doing that? Have you tried just to use the single pixel, after the sigma filter, as the test area? And still apply the same training area.

Response 6: I'm very sorry for the misunderstanding.In fact, in the previous part, we selected a small area for the experiment of optimizing filter parameters to provide reference for the subsequent data processing of the entire pearl river estuary basin.In the example application section, we did the filtering only once.

-----------------------------------------------------------------------------------------------------

Comment 7:

Minor:

Page 5 line 160: what do you mean with “two-phase”? Do you mean complex number?

Page 6 line 177: please change “radiation” into “radiometric”.

Page  7 line 208: Please change “non-uniform” into “noisy”.

Figure 13: Your search in the local area to evaluate if a target is a platform is a good thing. Misplacement is mostly due to errors in corregistration. Please add this in the manuscript.

Response 7: We are very sorry for our inappropriate writing. "Two-phase" means two different dates. “Corregistration” was indeed a very important factor that we added in the paper. Other inappropriate words have been corrected in the text.

-----------------------------------------------------------------------------------------------------

*Special thanks to you for your good comments. This is very helpful for our future research and writing.

Reviewer 2 Report

comments and suggestions for authors in the file "Review.docx"

Author Response

Dear professor,

Thank you very much for your careful review and sincere suggestions. We have revised the article according to your comments, the details are as follows:

Comment 1:

Page 5, section 3: Major parts of the section can me omitted, while some information is missing. Many researches have already explained how image correction, landmasking, speckle filtering and CFAR detection works. Mentioning the respective methods and including the respective references is sufficient as the explicit implementation details are not important for understanding the tuning of the CFAR control coefficient or for enabling the application of your tuning method. On the other hand, the exact definition of the conversion from the pixel’s digital numbers to the calibrated value is missing. This is important for motivating your choice for the optimal background distribution.

Response 1: Thank you very much for your comments. Some basic contents in this section are indeed mature work, but some reviewers think that some processes in this section is necessary, so we make some supplements and reservations. In addition, We calibrated the data using the SARscape module in ENVI 5.3 software.

-----------------------------------------------------------------------------------------------------

Comment 2:

Page 7, line 212: Why do you apply Gaussian distribution for background modeling? K-Distribution is meant to model the background clutter in a more accurate way on SAR intensity calibrated images and your main goal was the accuracy improvement of object detection.

Response 2:

Because K-distribution model is suitable for SAR data with medium resolution, the data used in this study is ScanSAR Wide mode data with low resolution. In addition, considering that the two-parameter CFAR algorithm adopted in this study has local adaptive ability, while K-CFAR algorithm cannot adapt to non-uniform SAR data.

-----------------------------------------------------------------------------------------------------

Comment 3:

Page 7, line 212: Page 9, lines 273 and 280: You write “n represents the total”, the total of what? Please provide more information about the calculation of E_max. How did you calculate E_t and E_b. In the current form I cannot follow what you did.

Response 3:

Thank you very much, this part of the expression is really not clear, appropriate.We have reorganized the calculation process of this part, and some modifications are shown below.

3.2.2 Estimation of the optimal false alarm rate control coefficient t based on maximum entropy

Segmenting the image of coefficient T distribution into target and background classes is a binary classification problem. In this study, the key to the problem is how to find an optimal control coefficient t from the set of coefficient T ()to segment it into two categories and achieve the best balance between target detection rate, false alarm rate and missed detection rate. Therefore, we use the maximum entropy index and a binary classification method to solve this problem [6, 7].

Under the maximum entropy principle, to predict the probability distribution of a random event, our prediction should meet all the known conditions, and we should not make any subjective assumptions about the unknown situation. This principle produces the most uniform probability distribution and the smallest predicted risk. Because the information entropy of the probability distribution is the highest at this time, we call this model the "maximum entropy model" [8-12]. Here, we consider all the T values as a probability distribution(Fig. 10). The control coefficient t value  that maximizes the information entropy of the target and background () is the optimal segmentation threshold, that is, the optimal false alarm rate control coefficient t. The calculation method is shown below.

Figure 10. Schematic diagram of optimal false alarm rate control coefficient t estimation based on maximum entropy. Eb represents the information entropy of background pixels, and Et represents the information entropy of target pixels.

According to the principle of information entropy, the probability distribution function of the possible results of each random variable  in the statistical frequency distribution of the coefficient T is , this satisfies [13] as follows:

.                         (3)

Therefore, the entropy of the coefficient T can be expressed by Formula (4) [14, 15]:

,                                 (4)

where , N represents the number of T values, and n represents the number of .

Similar to the above calculation method, when the control coefficient t is taken as (), the set of the entire coefficient T is divided into two sets: background and target, the information entropy Et of the target pixels, the information entropy Eb of the background pixels, and their sum Es  is shown in formula (5) - (7).

                          (5)

where ,  represents the number of T values in the background set, and  represents the number of  in the background set.

                          (6)

where ,  represents the number of T values in the target set, and  represents the number of  in the target set.

        (7)

Iterate over each of the coefficients  in the value range of () to get a set of the sum of the information entropy(). Then, according to the maximum value of it (), the optimal false alarm rate control coefficient t is obtained.

-----------------------------------------------------------------------------------------------------

Comment 4:

Page 9, section 3.3: Actually I don’t see a reason to calculate an optimal CFAR control coefficient. Your method for false alarm filtering would also work by using a much lower coefficient as the false detections of speckle would not be a problem here as it won’t occur on both images at the same location. Why do you not mention that you already published a very similar paper on this subject? (“Automatic Extraction of Offshore Platforms in Single SAR Images Based on a Dual-Step-Modified Model”).

Page 10, Figure 12: How did you calculate the TP, FN and FP rates?

Response 4:

Thank you very much for the comment.We added an explanation to the question. The value of t has a significant effect on the detection results. A larger t value reduces the number of false positives and increases the number of false negatives; conversely, a smaller t value reduces the number of false negatives and increases the number of false positives. Therefore, the t value must be adjusted repeatedly to achieve the best balance between the false positive rate and the false negative rate. Usually, t value is a parameter set subjectively based on experience or repeated experiments. Empirical parameters have certain reference value, but there are often differences between data from different sources or data from different coverage areas, so empirical parameters are not necessarily applicable. On the other hand, the experimental method usually can only select a parameter with relatively good detection results in the limited times of adjustment process, but it is also not necessarily the optimal parameter.

A lower false alarm rate control coefficient will detect more false alarm targets. We cannot assume that the false alarm target will never overlap in two scenes, and we must consider this possibility. At the same time, more false alarm targets will increase the computing time of the neighborhood analysis, which is not conducive to the automation of the algorithm.

In addition, our previous paper has some similarities with this paper, but the difference is that the previous paper adopted the SAR image data of medium resolution, and the key problem to be solved is how to eliminate the influence of ships in the SAR image data of medium resolution.

In this paper, the indicators used in precision evaluation are derived from previous published paper, which are cited. The accuracy of the figure is calculated according to the actual platform data in the experimental area.

-----------------------------------------------------------------------------------------------------

Comment 5:

Page 17, subsection 5.2, Figure 18: Your analysis of the reason for the missed detection is important. However, it is not enough to only investigate it qualitatively. 1) As the background is an area and not a line, Figure 18 would require a three dimensional plot. 2) The Gaussian distribution for the area should be plotted together with the intensity of the target pixel. 3) An overlay of the Gaussian distribution with the histogram of background values could reveal other reasons for the 2 missed detection, maybe it is due to your choice for using the Gaussian distribution as background model.

Response 5:

Thank you very much. That's a very good suggestion.This has deepened our understanding of offshore platform extraction research. We also modified the figure.

-----------------------------------------------------------------------------------------------------

Comment 6:

Page 11, section 4; Page 19, line 509-511: In its current form the research does not involve enough data to conclude that an optimal t value can be obtained by tuning the CFAR control coefficient using the maximum entropy method. A statistical analysis involving SAR scenes with multiple acquisition conditions is required. In the current version of your paper you do not even take the incidence angle into account. The following references provide you with more information about which conditions have an effect on the detectability and they also contain further interesting references.

Response 6:

Thank you very much. We are also aware of this problem. Therefore, in the application part, we compare the method proposed in this paper with the traditional method in terms of results and precision to illustrate the advantages of this method. In addition, we have referred to relevant literatures in the part of discussion, and added the influence of factors such as incidence angle and polarization mode on the extraction accuracy of offshore platforms into the discussion. This will undoubtedly make our study more complete and rigorous.

-----------------------------------------------------------------------------------------------------

Comment 6:

Minor:

Page 1, line 16: […] repeated adjustment of the during extraction […]

Page 2, line 59: SAR is not 100% independent from weather. Which other conditions do you mean? Please be more precise.

Page 3, line 120: In addition, in from the existing […]

Page 5, line 161: Which kind of masking do you mean? Landmasking? Please be more precise.

Page 6, Figure3: I suggest reordering the flow chart in a vertical way to improve readability.

Page 13, line 364: Now you mention your previous similar paper, but without reason.

Response 6:

Thank you very much. Some expressions are really inaccurate and inappropriate and had been corrected. And, some difficult to read charts, some of the content of the explanation are supplemented and revised.

-----------------------------------------------------------------------------------------------------

*Many thanks to you for your good comments. The questions were profound and detailed, and we benefited a lot from them.

Reviewer 3 Report

The manuscript "Offshore platform extraction using RadarSat-2 SAR imagery: a two-parameter CFAR method based on maximum entropy" presents a method to extract offshore platform targets from SAR images using a combination of CFAR algorithm and neighbourhood analysis (using images with different time acquisitions). An experiment was executed using 2 RadarSat-2 SAR images over the South China Sea.

The manuscript is well structured, but the methods presented are not new. Comparison of the proposed method with standard methods is necessary to proper illustrate the contributions of this manuscript. 

Please consider the following suggestion:

1) Compare the proposed two-parameter CFAR maximum entropy with the traditional CFAR algorithm. How they perform in AUC (Area Under The Curve)? What is the range of the coefficient t? How AUC changes with t and what is the best result for the proposed t0. How it compares with other CFAR maximum entropy methods already published?

2) The neighbourhood method is used to eliminate ship targets (non-stationary targets) using image overlap from two images taken 7 months apart. How this method performs with traditional CFAR? What is the impact of using it with the two-parameter CFAR maximum entropy? How the distance threshold of 150m was determined?

3) What is the estimated size for detected targets? What is the size of the missed offshore (target No. 37)?

Author Response

Dear professor,

Thank you very much for your careful review and sincere suggestions. We have revised the article according to your comments, the details are as follows:

Comment 1:

Compare the proposed two-parameter CFAR maximum entropy with the traditional CFAR algorithm. How they perform in AUC (Area Under The Curve)? What is the range of the coefficient t? How AUC changes with t and what is the best result for the proposed t0. How it compares with other CFAR maximum entropy methods already published?

Response 1: Thank you very much for your comments. Indeed, comparison can make our study more rigorous. Therefore, in the application part, the results and precision of this method are compared with the traditional method. The accuracy comparison and conclusion of the experimental part are as follows.

Table 4. Accuracy evaluation results of the automatic extraction method for offshore platforms.

Two-parameter CFAR based on   maximum entropy

Two-parameter CFAR

Optimal control coefficient t=0.7

empirical control coefficient

t=0.5

empirical control coefficient

t=1

Actual numberS

40

40

40

Total extractionN

42

41

40

NTP

39

38

37

NFP

1

3

1

NFN

2

0

2

TP

97.5%

95%

92.5%

FP

2.5%

7.5%

2.5%

FN

5%

0

5%

The evaluation results in Table 4 show that the two-parameter CFAR target detection method based on maximum entropy extracted 42 offshore platforms located in the Pearl River Estuary Basin of the South China Sea. Among these, 39 were correctly extracted, one was omitted and two were extracted incorrectly, achieving a TP rate of 97.5%, a FN rate of 2.5% and a FP rate of 5%. These results demonstrate that the proposed detection method has good performance for automatically extracting offshore platforms. Although certain false negative and false positive rates exist, their ratio is relatively low, which satisfies the extraction accuracy requirements of practical applications. The accuracy of the two empirical control coefficients is lower than that of the optimal control coefficient calculated based on the maximum entropy method, which also proves the superiority of the proposed method.

-----------------------------------------------------------------------------------------------------

Comment 2:

The neighbourhood method is used to eliminate ship targets (non-stationary targets) using image overlap from two images taken 7 months apart. How this method performs with traditional CFAR? What is the impact of using it with the two-parameter CFAR maximum entropy? How the distance threshold of 150m was determined?

Response 2: Because the offshore platform is a fixed target, in the traditional CFAR method, the fixed feature is also used to eliminate the moving ship target.The influence of neighborhood analysis on the method in this paper and the traditional method lies in the setting of search radius.This search radius can be considered as the deviation of the platform target at the same position in the two data. The larger the radius, some false alarm targets may be identified as platform targets. Since the length or width of the plane of the offshore platform generally does not exceed 150 meters, and the resolution of the data used in this paper is 100 meters, the radius threshold is set to 150 meters. This is not a fixed parameter and is usually determined based on the data used.

-----------------------------------------------------------------------------------------------------

Comment 3:

3) What is the estimated size for detected targets? What is the size of the missed offshore (target No. 37)?

Response 3: Because the data resolution of this paper is 100 meters, and the length or width of the offshore platform generally does not exceed 150 meters, the target detection results are generally points (that is, several pixels). Smaller platforms, such as unmanned platforms, may be less than 50 meters in size, have a simple structure, and the radar backscatter signal is weak. At this time, it may be a weaker target in SAR image, which may be missed in case of poor sea conditions.

-----------------------------------------------------------------------------------------------------

*Many thanks to you for your good comments. The questions were profound and detailed, and we benefited a lot from them.

Round 2

Reviewer 1 Report

I am glad to see that the authors did their best to implement the suggested changes. Therefore from my side, the manuscript is now ready for publication. 

Reviewer 2 Report

The authors link a series of methods for image processing, which are established in the research community for the task of object detection on SAR. Each of the methods have already been applied and published in the past for tasks similar to the ones they are applied for in this paper. However, they have not been combined in the way the authors do here and their combination establishes to opportunity to calculate an optimal detection threshold. Therefore, the scientific contribution is present. The authors responded to all comments from the last review round appropriately. The paper deserves publishing after minor revision, well done.

Minor:

Page 7, section 3.1.1: I am not familiar with this software and its modules. Further, at the end of a calibration process can stand an image in either amplitude or intensity representation. While amplitude images’ ocean background can better be modelled using Gaussian distribution, the intensity images’ ocean background is better modelled using k-distribution. Therefore, I find it important to add more information about the applied calibration method.

Page 8, section 3.2.1: Due to the smooth ocean background of low resolution images and also due to applied filtering, the Gaussian distribution might also be applicable to intensity images, so I agree to your answer to my second comment from last review and I recognize that you adapted the section. However, a reference would be appropriate.

Page 19, line 506: It seems the reference was not resolved correctly

Reviewer 3 Report

Thank you for considering the reviewers suggestions.

All my questions were satisfactorily addressed.